# Continuous evaluation of cost-to-go for flexible reaching control and online decisions

**Antoine De Comite** [ID][1,2,3], **Philippe Lefèvre**[1,2], **Frédéric Crevecoeur** [ID][1,2]*

**1** Institute of Neuroscience, UCLouvain, Louvain-la-Neuve, Belgium, **2** Institute of Information and Communication Technologies, Electronics and Applied Mathematics, UCLouvain, Louvain-la-Neuve, Belgium, **3** McGovern Institute for Brain Research, Massachusetts Institute of Technology, Cambridge, Massachusetts, United States of America

* frederic.crevecoeur@uclouvain.be

## Abstract

Humans consider the parameters linked to movement goal during reaching to adjust their control strategy online. Indeed, rapid changes in target structure or disturbances interfering with their initial plan elicit rapid changes in behavior. Here, we hypothesize that these changes could result from the continuous use of a decision variable combining motor and cognitive components. We combine an optimal feedback controller with a real-time evaluation of the expected cost-to-go, which considers target- and movement-related costs, in a common theoretical framework. This model reproduces human behaviors in presence of changes in the target structure occurring during movement and of online decisions to flexibly change target following external perturbations. It also predicts that the time taken to decide to select a novel goal after a perturbation depends on the amplitude of the disturbance and on the rewards of the different options, which is a direct result of the continuous monitoring of the cost-to-go. We show that this result was present in our previously collected dataset. Together our developments point towards a continuous evaluation of the cost-to-go during reaching to update control online and make efficient decisions about movement goal.

**Data Availability Statement:** We used our previous experimental data (https://datadryad.org/stash/dataset/doi:10.5061%2Fdryad.79cnp5hx8) to validate our model prediction. The github repo with the simulation codes is available at the

## Author summary

The way humans perform reaching movements is compatible with models considering that they result from the minimization of a task-related cost function. However, these models typically assume a cost function that does not change within movement, which is incompatible with experimental findings highlighting humans' ability to adjust reaching control online and change target flexibly. We hypothesized that this ability relied on continuous evaluation of the cost-to-go, which integrates task- and body-related information. We show that this model can optimally select and adjust control during movement in a way that reproduces human behavior in a set of tasks involving change in cost function and change in goal target. Our model predicted that decision-time to change target must be postponed when limb displacements and alternative rewards are smaller, which was borne out in our previous experimental dataset. To conclude, our model explains dynamic

following link (https://github.com/decomiteA/
CostToGoReaching/tree/main).

**Funding:** ADC and PL are supported by BELSPO
(Belgian Federal Science Policy Office) and the
European Space Agency. ADC is supported by a K.
Lisa Yang Integrative Computational Neuroscience
(ICoN) Postdoctoral Fellowship. The funders had
no role in the study design, data collection and
analysis, decision to publish, or preparation of the
manuscript.

**Competing interests:** The authors have declared
that no competing interests exist.

updates in reach control and suggests the cost-to-go as decision variable linking decision-making and motor control.

## Introduction

A commonly accepted hypothesis assumes that, when reaching towards an object, humans use a goal-directed feedback control policy tailored to the demands of the ongoing movement. This means that the observed reaching behavior is for instance tuned to the structure of the goal target [1–3], the reward associated with the movement [4–6], or the presence and location of surrounding obstacles [2,7,8]. This tuning does not only characterize unperturbed movements but also the way we respond to external disturbances. In this second scenario, studies have revealed the existence of flexible feedback control loops mediated through proprioceptive [1,7,9], visual [10,11], tactile [12–14], and vestibular [15,16] sensory inputs.

Besides this ability to cope with disturbances, recent findings reported that humans respond to a change in task demands by updating their controller during an ongoing movement. This was demonstrated in previous reports by varying the structure of the target during movement, a modification which is known to elicit different control policies when these targets are presented prior to movement onset [2]. Specifically, humans can exploit the redundancy of a target when there is multiple ways to attain a goal, as when reaching to an elongated object [1,2]. Even when the target changed after movement onset, participants were able to alter their control policy within 150ms [17,18] through regulation of the variance and displacement along the axis that was more or less constrained according to the suddenly changing goal. Similarly, external perturbations can evoke rapid decisions to switch to a new movement goal when there are multiple valid alternatives. The outcome of these decisions depended on information about the target reward, the motor cost, the state of the limb, and the perturbation magnitude, which clearly points to the use of a decision variable that considers both biomechanical factors and target-related costs [19–25].

We believe that this decision variable is associated with the *cost-to-go*, defined as the total expected cost to accumulate from any point in time until the end of the movement. It plays a central role in the optimal feedback control framework (OFC) [26,27], which posits that the closed-loop control policy underlying reaching behavior results from the minimization of the cost function defined by a weighted sum of motor errors and control-related costs [28]. Even though the OFC framework has been extensively used to explain flexible feedback control [2,29–32], it was mainly applied to static environments: that is, cost parameters have been typically considered fixed during a given trial. Therefore, a mechanism able to respond to these sudden changes in task parameters as observed in previous experimental studies is required. On the one hand, the cost-to-go depends on the target structure and reward through the cost function, as well as on the state of the limb including factors linked to the control of limb trajectory and potential disturbances. On the other hand, human behavioral responses to changes in movement goal described above depend on the same parameters. It is therefore natural to assume that the changes in reach behavior towards novel targets result from adjustments in control policy which reflects the dynamical estimation of the cost-to-go during movement.

To formalize these ideas, here we demonstrate that combining the OFC framework with a continuous evaluation of the task parameters, such as the target properties and the cost-to-go, captures participants' responses to dynamic changes in the task. We modeled online modifications in control by implementing a dynamical controller inspired by model predictive control [33,34], which was previously used to study sensorimotor control [35–37]. More specifically,

at each time step, the model extracts the relevant task parameters (target structures, locations and rewards), compares the cost-to-go associated with each option when multiple are available, and select the best action depending on the current task properties (e.g. target shape or lowest cost-to-go), see Fig 1 for a schematic representation of this model. Strikingly, this model predicts that the decision time to switch to an alternative target following an external disturbance must depend on the magnitude of that disturbance as well as on the relative rewards of the different targets. The reason is that the time required to displace the state of the limb to a region where the alternative goal becomes more attractive depends on the perturbation amplitude (smaller loads will take more time) and the relative target rewards (which modulates the size of the areas where each target is the most attractive), which is consistent with the diffusion models of decision-making [38]. Indeed, these models assume that decisions occur through the accumulation of a decision variable until it reaches a decision threshold. In the case of online control, we suggest that the decision variable is the cost-to-go and the threshold for the decision can be defined based on the difference of cost-to-go between alternative targets.

We reanalyzed our previous dataset and found clear evidence for a variation in decision time as predicted by the model. Altogether our developments support the theory that the brain monitors the target structure and the cost-to-go to operate dynamic and efficient adjustments of control. Our results also make a direct link between current theories of sensorimotor control and decision-making by suggesting that the cost-to-go is the decision variable used in the brain when we move in a dynamical context.

## Results

We extended the optimal feedback control framework by adding a continuous tracking of the task parameters (targets locations and structures, rewards), and limb trajectory such that the control policy can be adjusted in dynamical contexts (changes in target structure, unexpected perturbations). We implemented a hierarchical controller, where a high-level controller adjusted the low-level Linear Quadratic Gaussian controller (LQG) to task demands, by continuously evaluating and comparing the current task parameters and potential alternative options (Fig 1, the gray and black loops respectively represent the high- and low-level controller, see Methods for more details). For instance, when the goal target switched from a square to a rectangle during a movement, the high-level controller observed that the target shape has changed and adjusted the low-level control policy to the new target structure for the remaining

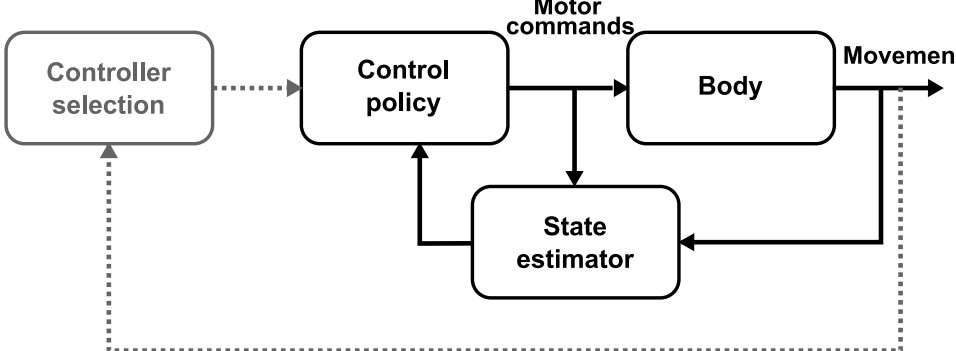

**Fig 1. Model architecture.** The feedback-based controller from the optimal feedback control framework (rightmost loop, black) is augmented by an additional loop (leftmost loop, gray) that selects the controller as a function of task parameters and state estimation. This additional loop is responsible for controller adjustments during movement.

part of the ongoing movement. Similarly, when the various alternative targets differed by their location and reward, the higher level of the controller compared their respective cost-to-go, which depended on task-related parameters such as target reward or the state of the system, and adjusted the low-level control policy such that the reach goal was the lowest cost-to-go target. In this implementation, the cost-to-go associated with each alternative were evaluated at each time step such that the feedback gains of the low-level controller can be adjusted to the option associated with the lowest cost-to-go. To validate this model, we reproduced a series of experimental studies involving change in target redundancy during movement and online decisions to change in the movement goal.

## Online changes in target structure

In the first behavioral study reproduced here, healthy participants were instructed to perform reaching movements to a target which could suddenly change from a narrow square to a wide rectangle or vice versa after movement started [17]. Lateral step disturbances were used to elicit feedback responses and reveal potential updates in control. Participants were able to adjust the way they responded to the disturbance within movement according to the changes in target width on average 150ms after the target switched. We simulated this experiment and used the 150ms delay as a parameter to update the control policy based on the actual target structure, which differed from the 50ms delay in the feedback loop associated with physiological delays of long-latency responses [39,40].

In our simulations, the model captured the fact that participants let their hand deviate towards more eccentric locations along the redundant axis when the target switched from a narrow square to a wide rectangle (Fig 2A, dotted red line versus full red line), and the opposite behavior was reproduced when the target switched from a wide rectangle to a narrow square (Fig 2A, dotted blue versus full blue lines). An alternative hypothesis would be that, in the absence of a module dedicated to the detection of the alternative options and to the alteration of the controller structure, the expected behavior should correspond to the one observed when reaching for the initial target (narrow square and wide rectangle for the narrow-to-wide and wide-to-narrow conditions respectively). Under this assumption, we would observe a larger end-point error for the wide-to-narrow conditions (full line blue, Fig 2A) and a larger corrective response for the narrow-to-wide condition (full line red, Fig 2E), which is inconsistent with the experimental data [17]. When the target structure changed during movement, lateral hand deviations induced by the mechanical disturbance (Fig 2B) were similar to what was experimentally observed. Indeed, participants' hand ended at more or less off-centered positions in the narrow-to-wide or wide-to-narrow conditions, respectively. The forward movement, aligned with the main reaching direction, was not influenced by the change in target structure while the end-point variances along the x-axis clearly depended on the condition (Fig 2C) as we reported experimentally. The transverse velocity (aligned with the x-axis) was also clearly modulated by the experimental condition (Fig 2D), which resulted from the differences in motor commands acting along that transverse direction.

Our model also reproduced the increase in the intensity of motor commands that can be qualitatively compared to the increase in EMG activity measured in the muscles stretched by the mechanical perturbation. Even though our implementation was not designed to fit EMG data precisely, we could compare the relative changes (increase or decrease) in motor commands along the x-axis (Fig 2E) with the modulation of muscle responses following wide-to-narrow and narrow-to-wide switches. In the wide-to-narrow switch (dashed blue line), the model displayed an increase in motor commands consistent with the increase in the electromyographic (EMG) activity of the stretched muscle. Similarly, in the narrow-to-wide

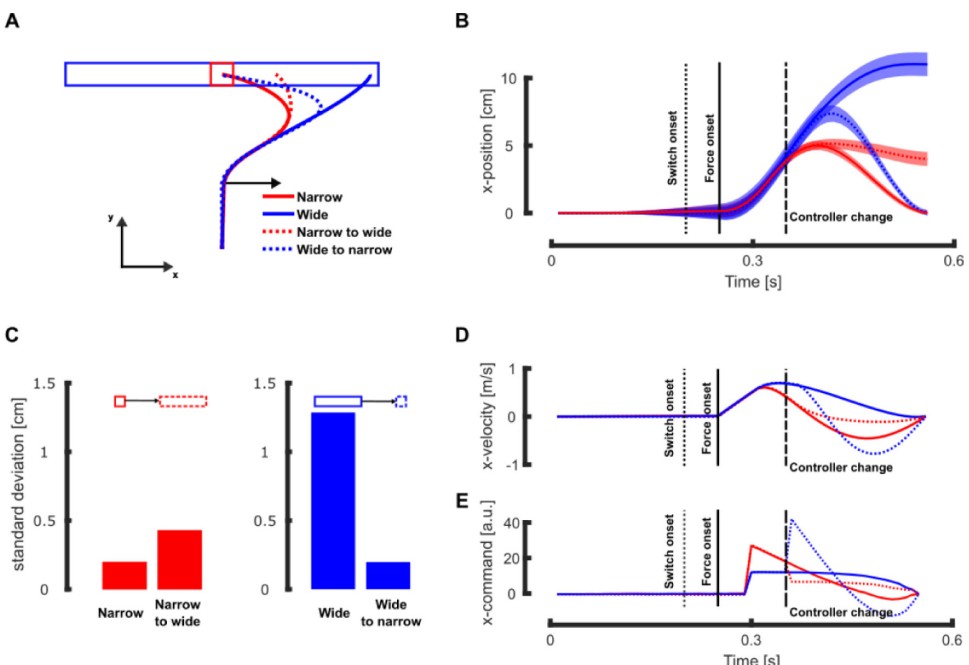

**Fig 2. Simulation results: narrow and wide targets. A** Mean reaching traces in presence of a rightward mechanical perturbation (represented by the black arrow) for trials initially directed towards a narrow (red) or wide (blue) target. The full and dashed lines represent the trials with and without switch in target structure, respectively. **B** Mean and standard deviation of the x-position for the different target conditions. Time is aligned with movement initiation and the vertical dotted, full, and dashed lines respectively represent the target switch onset, the force onset, and the time at which the controller was updated. **C** Simulated end-point variances along the x-axis in absence of mechanical disturbances for trials without (left bar plot) and with change in target structure (right bar plot). The left panel represents the switch from square to rectangle and the right one the switch from rectangle to square. **D** Mean traces of the transverse velocity in the different target conditions. **E** Mean traces of the x-motor command for the different target conditions. For comparison with experimental data, the panels A, B and C correspond to the Fig 3 in [17] and the panels D and E correspond to the Fig 4.

condition (dashed red line) the model produced a decrease in motor command also compatible with our previous experimental findings [17].

Next, we confronted the model with our more recent experimental study in which we reported that online adjustments in the control policy were sensitive to dynamical factors such as the rate of change in target width [18]. For this purpose, we simulated movements towards an initially wide target (black in Fig 3A), which could continuously become narrower during movement with different rates of change. As in our experimental study, we considered three different changes in target width: a sudden switch to a narrow target (*switch* condition, magenta in Fig 3A), a fast continuous decrease in target width (*fast* condition, blue in Fig 3A), and a slow continuous decrease in target width (*slow* condition, green in Fig 3A), see Methods for the detailed implementation of these conditions within our model.

Our model reproduced the behavioral observations for each target condition (Fig 3A). The end-point distribution along the x-axis, which indirectly quantifies how much the perturbation has been corrected, clearly depended on the target condition. The mean end-point position was more eccentric for less constraining target conditions. The model also captured how the target condition influences feedback responses to mechanical perturbation, leading to changes in lateral hand deviation and velocity along the lateral axis (resp. Fig 3B and 3D). Changing the target structure did not affect the movement along the main reaching direction, consistent with our experiments and the minimum intervention principle that is a known feature of

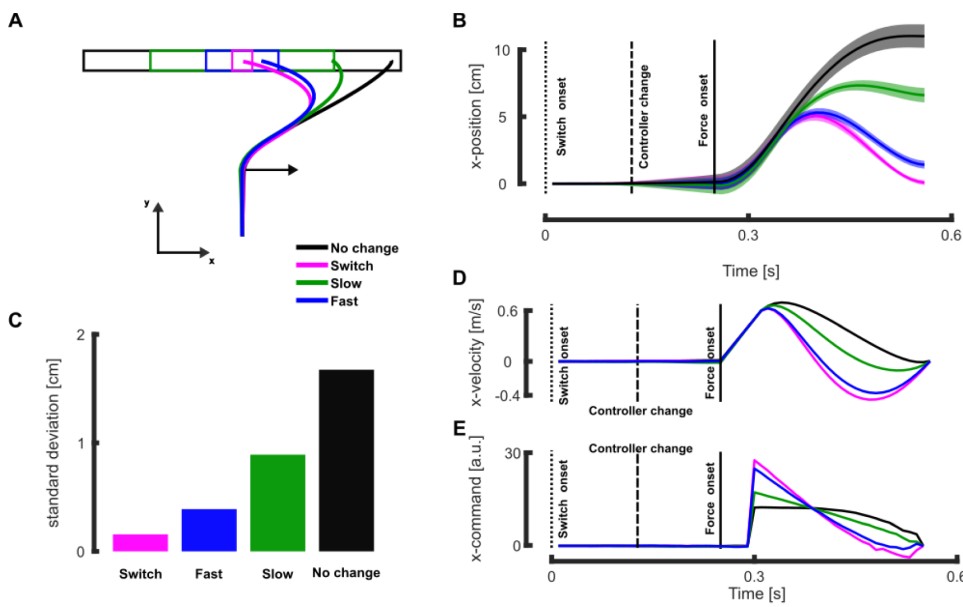

**Fig 3. Simulation results: Continuous change in target width. A** Mean reaching traces in presence of a rightward mechanical perturbation (represented by the black arrow) for the different target conditions in the absence (black) or presence of target change (green: slow continuous change, blue: fast continuous change, and magenta: instantaneous change). **B** Mean and standard deviation of the x-position for the different target conditions. Time is aligned with movement initiation and the vertical dotted, dashed, and full lines respectively represent the target switch onset, the time at which the controller was updated, and the force onset. **C** Simulated end-point variances along the x-axis in absence of mechanical disturbances for the different target conditions. **D** Mean traces of the transverse velocity for the different target conditions. **E** Mean traces of the x-motor command for the different target conditions. For comparison with experimental data, the panels A, B, and C correspond to the Fig 2 in [18] and panels D and E correspond to the Fig 3.

human feedback control strategies [27]. We also captured the patterns of activations in the stretched muscles reported in the experimental study using the motor commands along the x-axis. We observed that this motor command was modulated by the target condition and that it even scaled with the rate of change in target width such that the more constraining changes in target structure were associated with larger motor command (Fig 3E). The fact that different changes in target structure have different impact on behavior, rules out the alternative hypothesis of an ad hoc adjustment of the controller that would be determined before movement started and independent of the rate of change in target structure.

Importantly, the model captured the fact that the target structure modulates both the mean behavior and its variability. When the target width was reduced during movement, the movement variability along the transverse axis decreased as well (see Fig 2B, dotted vs full blue and Fig 3B), the reverse effect was observed when the target width increased during movement (Fig 2B, dotted vs full red). This modulation of the behavior variability is further illustrated by the end-point distribution, showing a clear dependency on the target condition for the unperturbed movements (Figs 2C and 3C). This indicates that the controller fully exploited the target redundancy to control movement, similarly to what was reported experimentally.

Together, these results demonstrate that the recursive computation of feedback gains that we proposed to model tasks involving online alteration of the task demands was able to reproduce (1) online updates in the structure of the controller, (2) modulation according to dynamic factors such as the rate of change in target width, and (3) exploitation of target redundancy visible in the end-point coordinates and variances. This recursive implementation of motor

control which integrates changes in the environment can be further extended to study participants' reaching strategies in presence of multiple alternative goals as is presented below.

## Online motor decisions between alternative motor goals

Decisions to commit to a specific target are influenced by a wide range of factors, such as the reward of each target [21], the biomechanical costs [19], or even the task constraints [24]. Here we formulate the hypothesis that the selection of a target (offline or during movement) directly results from an evaluation of the cost-to-go, and that online changes in movement goal occur when the estimate of the cost-to-go for an alternative target becomes more attractive than the one associated with the initial decision. Such changes can occur for instance when a perturbation pushes the hand towards an alternate goal or when the respective reward of the different targets suddenly changed during movement.

To integrate the reward of each option in the cost-to-go, we additively biased the cost-to-go of less rewarding targets with a positive constant (see Methods for more details). Biasing the cost-to-go penalized these targets and favored decisions to reach for targets with higher rewards. To validate this implementation, we simulated our previous experiment in which participants had to reach for any of three alternative targets that could be associated with different rewards [21]. We show in Fig 4 that our implementation was able to predict the patterns of online motor decisions, in particular the statistical dependency on the frequency of switch on both the reward of the alternate target and on the perturbation magnitude.

These simulations reproduced the increase in the frequency of switches to the lateral targets after the application of a mechanical disturbance when the load was larger or when the lateral target had a comparable reward [21,25]. So, the model captures the behavioral observation that these two parameters played a significant role in the statistical model of target switches (Fig 4A and 4B). To further quantify the impact of the reward distribution on switching behaviors, we simulated a wider set of reward distributions and observed that, the larger the difference between the central and lateral targets (i.e. the more the central target was rewarding compared to the other two), the smaller the frequency of reach to the lateral targets (Fig 4C).

Importantly, the commitment to reach a target was not imposed and the controller could select to reach whichever target was more profitable according to the current estimated cost-to-go. In the experiment that we simulated, the selected target prior to the onset of mechanical perturbation was the central target as it was significantly closer to the hand than the other two (see Fig 4D–4F, the magenta line lies below the other two before force onset). The mechanical perturbation altered the cost-to-go associated with each target by displacing the state of the system physically, which impacted the biomechanical factors and the cost of correcting for the perturbation, to a point where even less rewarded targets could become more profitable in comparison with the larger reward of the center target that requires a larger and costlier corrective action (see the small bump at the dashed line in Fig 4E). In the trial illustrated in Fig 4F, the controller selected a different target with lower cost-to-go value (the red arrow in Fig 4F captures the switch to the right target), which was less rewarding but also less effortful. Since we did not impose the goal target at any time during movement we observed, for a same experimental condition (reward distribution and force amplitude), different behaviors (see Fig 5A, different colors for a same panel). This flexibility to decide could not be reproduced if the controller only considered a single reach target for the current movement.

The implementation that we proposed to simulate online motor decisions can also be used to simulate movements towards a redundant target such as a rectangular target. To show this, we extended our simulations to other experimental protocols investigating online motor decisions towards redundant targets associated with a non-uniform reward distribution along the

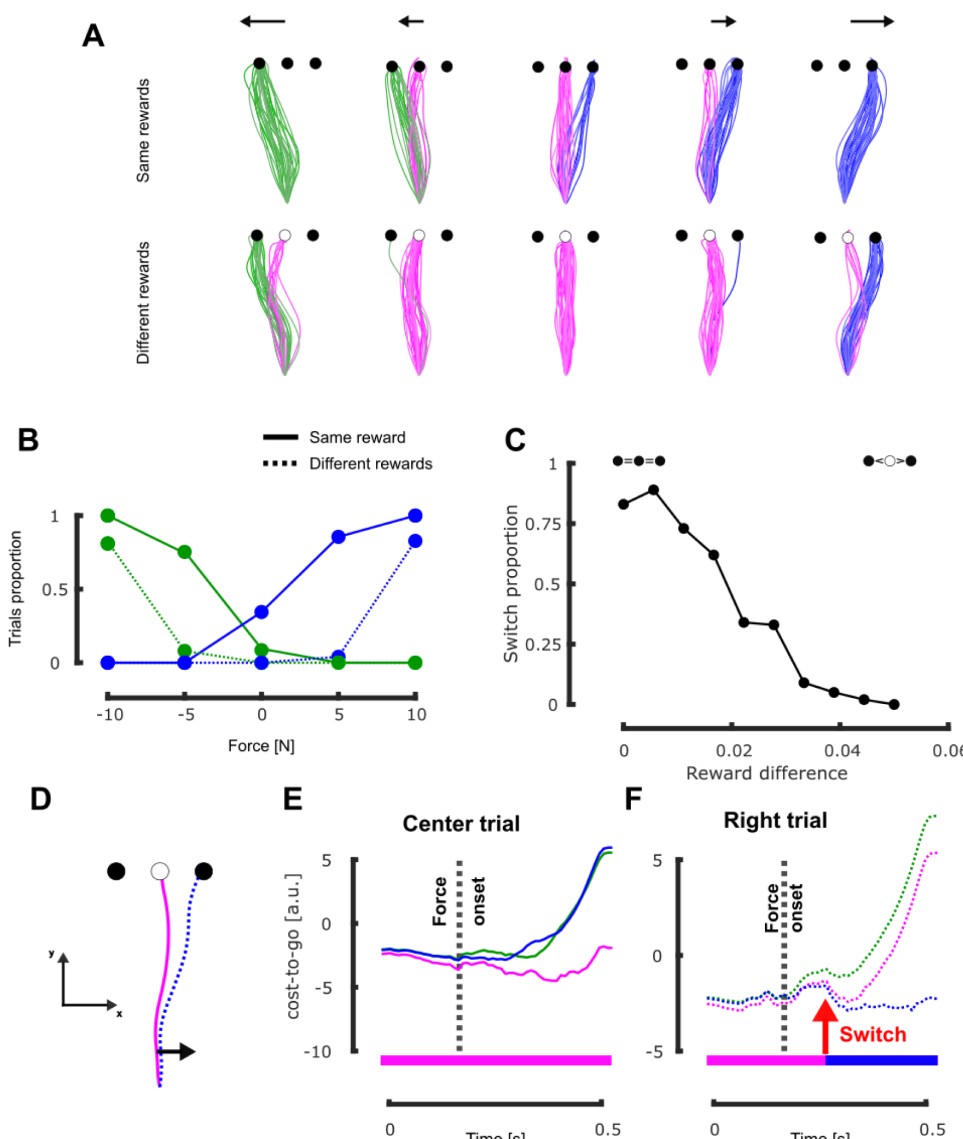

**Fig 4. Simulation results: Online motor decisions. A** Individual simulated hand traces in presence of multiple alternative targets (green, magenta and blue traces represent trials reaching the left, center and right target respectively). Each row corresponds to a different reward condition (top and bottom for same and different rewards, respectively) and each column represents a different force level for the mechanical disturbance (-10N, -5N, 0N, 5N, and 10N from left to right). **B** Proportion of the trials that reached the lateral targets (green and blue lines for left and right targets, respectively) as a function of the intensity of the mechanical perturbation. Full line represents the same reward condition and dotted line represents the different rewards condition. **C** Proportion of trials that reached the rightward target in presence of a rightward disturbance (5N) as a function of the difference in reward between the central and the right targets. Positive difference values favor the central target which was more rewarding than the lateral ones. **D** Hand traces for the illustrated example of the relationship between cost-to-go function and behavior, corresponding to the fourth condition of the second row of panel A (different rewards, slight rightward perturbation). The full magenta line represents a trial where participant's hand reached the central target (corresponding to the graph of panel **E**) and the dotted blue line represents the one that reached the right target (corresponding to the graph of panel **F**). **E-F** Representation of the cost-to-go values associated with each target (green = left, magenta = center, blue = right) for trials in the different rewards condition and rightward mechanical perturbation. The panel **E** represents the cost-to-go values for the trial that ended at the central target and the panel **F** represents those for the trial that ended at the right target. The red arrow captures the time at which the right target became the goal target for the right trial and the rectangular insets at the bottom of the panels represent the target associated with the lowest cost-to-go at each time. Time axis is aligned on movement onset. For comparison with experimental data, see the Fig 4 in [21].

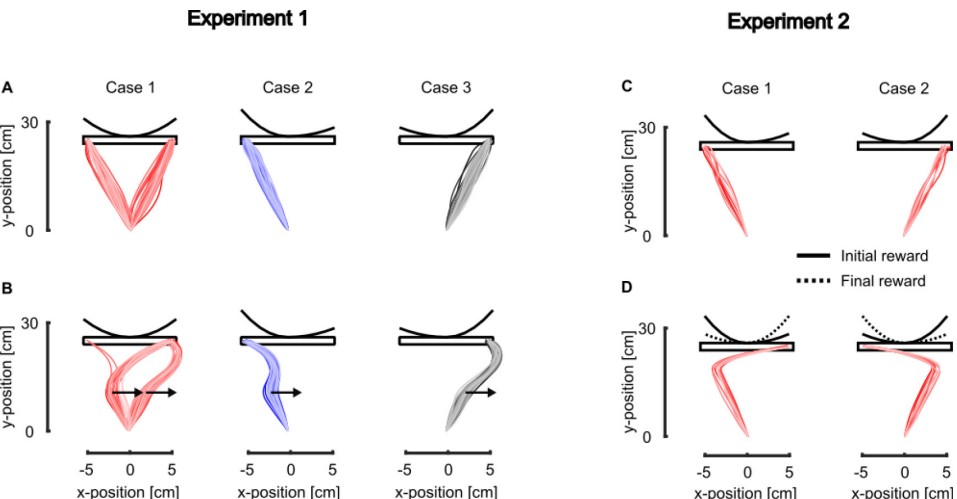

**Fig 5. Online motor decisions for rectangular targets. A** Individual simulated hand traces for the symmetric (Case 1) and asymmetric (left bias and right bias for cases 2 and 3, respectively) reward distributions. The reward distributions along the x-axis are represented above the targets. **B** Individual simulated hand traces for the three different reward distributions in presence of a rightward mechanical perturbation. **C** Individual simulated hand traces for the biased distributions (left and right bias represented by cases 1 and 2, respectively). **D** Individual simulated hand traces in presence of a switch in the reward distribution (from the full line to the dotted line) for both initial reward distributions. For comparison with experimental data, see the Fig 2 in [20] for the panels A and B and the Fig 3 in [23] for the panels C and D.

redundant axis [20,23]. In their recent study, Cos and colleagues instructed participants to perform reaching movements towards a rectangular target that had non-uniform reward distributions [20]. We simulated a similar experiment and considered three different cases for the reward distribution, as in the original experiment. In the first case, the reward distribution was symmetric with respect to the center of the target (Fig 5A, Case 1). The two other cases considered asymmetric reward distributions with either a leftward (Fig 5A, Case 2) or a rightward (Fig 5A, Case 3) bias.

Our model captured participants' reaching behavior in both unperturbed and perturbed conditions. In the unperturbed condition, the model predicted that participants were biased towards locations associated with the highest reward: both extremities on the target in the symmetric condition (Fig 5A, Case 1) or the extremity associated with the largest reward in the asymmetric conditions (A, Cases 2 and 3). Interestingly, it also predicted the *changes of mind* observed in the original study by Cos and colleagues as can be observed in the symmetric reward conditions in presence of rightward mechanical perturbation (Fig 5B, Case 1), where some trials initially directed to the left end of the target ended up at the right end after the application of the perturbation.

We can combine the bias of the cost-to-go function with an online change in the reward distribution itself to model human behavior in tasks involving online change in task demands and online motor decisions between multiple targets. In a recent study, Marti-Marca and colleagues [23] instructed participants to reach for a target represented as a wide rectangle aligned with the x-axis which has a non-uniform reward distribution along this axis. In the present study, we only considered the reward distributions schematically represented above the targets in Fig 5C. As participants were reaching towards this target, the reward distribution could suddenly switch from one condition to the other (i.e. the reward bias moved from left to right or vice-versa). Our model predicted that the online changes in reward distribution impacted the reaching behavior similarly to what has been reported experimentally [23]. Indeed, in Fig 5D

we reported that the trials that were initially targeted towards the end of the target associated with the high initial reward (full line traces in Fig 5D) suddenly redirected towards the other end after the reward distribution was changed (dotted line traces in Fig 5D). This result further demonstrates the ability of our model to predict the outcome of online motor decisions in dynamic contexts.

### Experimental evidence for continuous monitoring

Our strongest claim is that the nervous system evaluates the cost-to-go in a continuous fashion. Our interpretation is falsifiable in two ways: a first alternative hypothesis could be that the cost-to-go is not recalculated online following changes in task demands and that there is no module able to adjust the control policy online. This first hypothesis could be rejected based on the experimental observations described above. The second alternative hypothesis would be that the evaluation of the cost-to-go that underlies motor decisions is not continuous but instead is triggered by discrete events such as perturbations or visual cues. We know that, in the experiment presented in Fig 4 decisions to switch target depended on the occurrence of an external perturbation, thus the minimal candidate model that does not feature a continuous evaluation of the cost-to-go is a discrete evaluation of this quantity, potentially triggered by the occurrence of the load disturbance. There is a testable difference between such an event-triggered switch and a continuous monitoring: in the case of a discrete switch, for a given perturbation magnitude, the decision must be taken at the same time irrespective of the different target rewards. In contrast, the continuous model features the possibility that decisions are taken at different times. The reason is that, in the condition where the lateral target has lower reward, it requires a larger hand displacement before it becomes more attractive, thus the decision to switch target is postponed until the hand has travel a larger distance, if it does. Similarly, for different perturbation magnitudes and a fixed reward, the same reasoning suggests that decisions to switch can be taken faster following larger perturbation magnitudes. Thus, contrary to a discrete switch hypothesis without ad hoc tweaking of the decision time, a continuous evaluation implies that decisions can be taken at different times as a consequence of the continuous evolution of the limb trajectory.

This effect was observed both in simulations, as expected, and in our previous dataset [41]. In the simulations a clear delay in decision time was reproduced when lowering the reward of the lateral target gradually (Fig 6A). In addition, we reanalyzed our previously published data in search for a delaying of the decision time dependent on target reward and perturbation magnitude. We compared for each combination of perturbation direction and reward distributions the time at which trials reaching the central and lateral targets could be distinguished. We used the receiver-operator characteristic (ROC) technique to determine when these trials could be distinguished (see Methods). We found that this time onset was modulated across reward conditions and perturbation directions (Fig 6B). Strikingly, we observed, exactly as in the model simulations, that the onset of decision occurred earlier when all the targets had the same reward (leftward perturbations 147 vs 190ms Fig 6C vs 6D and rightward perturbations 163 vs 186ms) than when the central target was more rewarding. Similarly, we found in the same dataset that the amplitude of the perturbation influenced the decision time: smaller perturbation amplitude resulted in longer decision time (0.161s (small) vs 0.147s (large) for the leftward perturbations, Fig 6E vs 6F, and 0.191s (small) vs 0.163s (large) for the rightward perturbations). This decision time is also impacted by the amount of motor noise (simulations not shown), which increases the decision time at higher level of noise but does not affect the difference between both reward conditions. Even if our model was able to qualitatively predict the modulation of decision time with the conditions, the predicted time does not match the

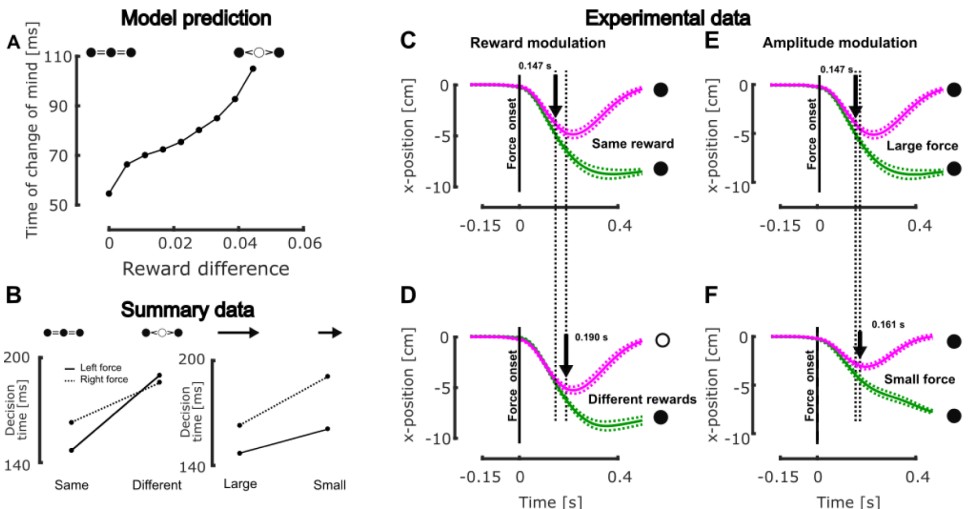

**Fig 6. Behavioral evidence of the modulation of decision time. A** Model prediction of the modulation of the time of change of mind as a function of the difference between the central and the lateral targets. In presence of larger reward difference (when the central target has a larger reward than the other two) the model predicted longer decision time. **B** Summary of the experimentally measured decision time for constant force and various reward distributions (left) and constant reward distribution and different force levels (right). The full and dashed lines represent the trial with a leftward and rightward perturbation, respectively. **C** Group mean and SEM hand traces along the x-axis for reaching performed in presence of three alternative targets and leftward perturbations. All the targets had the same reward, the magenta trace captures trials that reached the center target while the green trace captures those that reached the left one. The vertical dashed line represents the onset of mechanical perturbation and the black arrow the onset of change of noticeable differences between the two targets. **D** Same as **C** in the condition where the central target was more rewarding than the lateral ones. **E** Same as **C**. **F** Same as **E** in presence of small leftward perturbations instead of large ones.

one from the data probably because the model does not implement all the mechanisms underlying the control of movement such as short-latency reflexes for instance. We indeed designed the model such that it can qualitatively, rather than quantitatively, reproduce and predict experimental results (see the standardized parameter design in Methods).

In all, these results demonstrate that our hierarchical implementation of OFC which leverages the cost-to-go function is able to capture (1) reaching behavior in presence of online changes in target structure and (2) the decision to switch to an alternative reach goal in presence of change in task in a way that depends on the perturbation and reward distributions. Importantly, our model predicted that the time required to switch to a novel goal should depend on the amplitude of the perturbation and on the reward distribution since the cost-to-go was defined based on the external target reward and the time-varying state of the limb. This prediction was verified by reanalyzing our previous dataset.

## Discussion

In the present work, we demonstrated that human reaching movements in dynamical environments could be captured by an optimal feedback control (OFC) model modified to continuously track the target structure and the cost-to-go associated with the current and alternative goals. This modification enables online adjustments in control in response to change in target shape and decisions to reroute the ongoing movement to a novel goal. We implemented a recursive controller which was able to reproduce online motor decisions that depend on the state of the system, the presence of external disturbances, and the respective reward of the different available reach options. It also prompted us to reanalyze previous data in search to highlight a dependency of the decision time on target reward that was waiting to be discovered.

We combined a decision module, based on the continuous observation of the target structure and on an estimation of the cost-to-go, with the OFC framework to accommodate the experimental observations made about the dependency of reaching behaviors with sudden changes in goal. The addition of this decision module was necessary since the native OFC implementation traditionally assumes that the cost function, and therefore the control policy, is fixed during movement. This assumption was incompatible with the experimental results captured in this study. Here, we used a hierarchical controller which sequentially monitors the task requirements (target structure, reward), then selects the motor commands associated with the best option to simulate reaching behaviors in these changing conditions. The present work used this hierarchical organization of the controller to capture the decision processes occurring during movement, while such organization was previously used to cope with the curse of dimensionality inherent to the control of neuromuscular models [41–43], to define a hierarchy of goals [44], or to bring together control and adaptation during reaching and locomotion [45,46]. More specifically, the higher level of our model combines the notions of dynamical updates in the controller with the estimates of the cost-to-go to evaluate and compare concurrent options, while the lower level of our model implemented as a linear-quadratic-Gaussian (LQG) controller [2,27], executes the control policy associated with the current selected option.

Recursive controllers, often implemented in the Model Predictive Control (MPC) framework which assumes that the control problem is recursively solved during movement [34,47], have previously been used to explain experimental features that were not captured by other models: the temporal evolution of feedback gains during reaching movement [48], or the modulation of the movement duration of reaching movements in presence of perturbations [33,49,50]. In a recent work, the MPC framework was combined with an impedance controller to demonstrate its ability to handle nonlinear dynamics and changing environments (i.e. force fields) during movement [51]. Thanks to the recursive computations of the feedback gains inherent to this framework, Model Predictive Control was also used to investigate the control horizon in sensorimotor control [35–37]. Here, we took inspiration from this MPC framework to combine the recomputation of the control policy with the continuous evaluation of the cost-to-go function through a generalization of a previous implementation [25] proposed by Nashed and colleagues by integrating the reward within the cost-to-go function by a positive cost bias for the less rewarding options. Our implementation is dynamical in the sense that it computes and compares the cost-to-go of the different options at each time step using the same current state to predict the switch in behavior at any time. This allows us to reproduce the shift in behaviors and decision processes reported in multiple experimental studies involving reward [20,21,23]. In the presence of higher reward, participants will be more prone to take actions associated with higher motor costs which are for instance characterized by higher velocities and feedback responses [5,21,52].

Our main contribution is to provide support for the hypothesis that the cost-to-go values of the different alternatives are continuously compared, and the motor command associated with the lowest value is selected, which allows formulating a dynamical implementation of the distributed consensus model during movement [53]. The basic premise of this model is that decisions between multiple motor actions result from the integration for each option of high-level factors and low-level biomechanical costs, which both bias the competition between options with reciprocal interactions during the process leading to a motor decision [19,54]. In other words, the decision to take an action does not only depend on its outcome (e.g., its associated reward), but also on the cost of selecting it (e.g., its biomechanical cost). The cost-to-go function encapsulates both the impacts of action outcome and that of motor cost incurred to that action. Importantly, the fact that a decision variable depends on these components was a

constraint dictated by previous experimental work. Interestingly, the concomitant influence of reward and cost parameters was not only observed in tasks involving distinct reach options (Fig 3) but also in tasks involving a redundant target (Fig 4). Our algorithm does not require selective adjustments to explains these two phenomena since the cost-to-go function directly derivates from the cost matrices that feature or not a redundant dimension in movement goal. The model is therefore quite general, and flexible enough to handle a broad range of parameters linked to movement control.

We bridged an important gap between current models of motor control and decision making by introducing the cost-to-go as a decision variable. We selected the cost-to-go as a decision variable because of its theoretical groundings in the dynamic programming solution of the optimal control problem [55]. It was indeed used to derive locally and globally optimal solutions of the control problem through stochastic optimal control [27,56] and reinforcement learning, where it is often referred to as value function [57,58]. The implementation presented in this paper allows to go beyond standard static formulation of OFC models with fixed parameters, and explicitly integrate the reward, changes in target structure and multiple targets in the cost-to-go. Importantly, the paradigms studied here differ from those of previous studies which investigated how humans performed reaching movements in presence of uncertainty on the goal target [59–62]: multiple potential targets were presented before movement and the correct one was cued once the participants had started moving. Most of the behaviors observed in these go-before-you-know paradigms can be reproduced by averaging the control policies or target locations associated with the different options [62–65], simulations not shown. Thus, from a computational perspective this problem does not require to update the controller online while changes in target structure during movement and the decision processes occurring when multiple options are available throughout movement do require such update. In the current version of our model, we are not able to capture all the experimental conditions in the Wong and Haith [61] or Alhussein and Smith [64] studies, however a controller performing dynamic updates of its policy based on continuous tracking of the cost-to-go, such as the one developed in the present study, can also handle the problem of reaching to uncertain targets.

Our approach has some limitations. The cost functions were independently defined for each experimental condition using a different set of parameters, rather than by a constant set of model parameters that could flexibly express multiple cost functions and from which different behaviors could emerge. This limitation is tightly linked to our choice to model the controller with a linear-quadratic-Gaussian regulator, which constraints the cost function to a quadratic function of the state and action, this problem is recurrent in most of the studies that use the Optimal Feedback Control framework to simulate reaching movements [27,30,66,67]. Another important limitation of our model is related to our choice of time horizon for the controller formulation. Indeed, we chose to use a finite horizon controller across all the experimental paradigms, which induces that our model cannot predict the modulation of movement duration with parameters like reward [5,68–70] or required accuracy [71]. Modulations of movement duration cannot naturally emerge from our model and the incorporation of updates in movement times requires further developments either by considering a receding horizon [72] or an infinite horizon [73,74]. We did not include signal-dependent noise in the results presented in the Results section since its contribution does not affect our conclusion. We detailed how to generalize our model to signal dependent noise and reproduced the main results in the S1 Text. Finally, we did not model the documented impact of reward on the modulation of movement velocity and feedback gains [5,21,52,68,75]. Previous works attributed these dependences on reward to the value of time and its influence on the perception of reward [69,76,77]. This hypothesis was confirmed by studies that brought together these notions with optimal control [72,78,79]. Recently, the influence of uncertainties about the

environment [80,81] and reward [6,21] on reaching behavior has been reported. To account for the modulation of movement velocities and feedback gains related to temporal factor or to the use of a robust strategy [81], a fuller implementation requires an additional module which not only selects the current target or controller, but also modifies the control gains according to the vigor or robustness of the intended movement.

The dynamical tracking of the cost-to-go enabled the reproduction of the behavior of healthy adults reported in multiple previous studies [17,18,20,21,23–25]. The neural structure supporting these within-movement adjustments has not been experimentally identified, but we hypothesized that the basal ganglia are likely to be involved. Indeed, this structure is often associated with the evaluation of the cost of a movement [82–84] which results in the invigoration of movements often associated with the presence of reward [5,21,52,68]. It is conceivable that this structure is also involved in the representation of the structure of the goal as well as in the target selection. Besides their potential implication in the evaluation of movement costs, the basal ganglia also play a central role during decision-making. Indeed, seminal models suggest that the basal ganglia compare and select the actions to be performed [85,86]. The exact role of the basal ganglia in decision-making is nonetheless not yet fully understood [84,87,88]. Their implication in the online adjustments process we reported in the present work are likely to be combined with other brain areas [89–92].

Together our modeling results expand the applicability of optimal feedback control framework to tasks involving dynamic changes in parameters during movement and formalize the distributed consensus model suggested by Cisek [53] to the control of reaching movements. By bridging together motor control and decision-making, the present work opens new modeling and experimental perspectives to investigate movement control in complex environments. From a normative perspective, control models must be enriched with online estimate of movement costs and updates in control. From the perspective of neural implementation, it implies that humans do not sequentially decide, plan, then act. Instead, these operations seem imbricated in a general, continuously evolving process. Finally, we believe that the similarities between the theoretical framework in which we grounded our model and that of reinforcement learning suggest direct theoretical developments to further challenge our understanding of human sensorimotor control in the context of artificial systems designed to mimic human behavior: not only reward signals must be used to train a controller, but control signals must also be used to train and form an estimate of the expected reward. Exploring how this can be done in neural systems is an exciting challenge for prospective studies.

## Methods

### General model

We considered the translation of a unit point mass (m = 1kg) in the horizontal plane in presence of three forces: a viscous force proportional and opposite in sign to the velocity, a controlled force $F$, and an external force $F_{ext}$. The controlled force was represented by a first-order low pass response of the control vector to approximate non-linear muscle dynamics. The controlled system was described by the following set of continuous differential equations that characterized movement along the x- and y- axes independently

$$m\ddot{p}(t) = -G\dot{p}(t) + F(t) + F_{ext}(t) \tag{1}$$

$$\tau\dot{F}(t) = u(t) - F(t) \tag{2}$$

$$\dot{F}_{ext}(t) = 0 \tag{3}$$

where $m$ is the mass, $p(t)$ is the two-dimensional vector of the point mass coordinate in the plane, $G$ is a viscous constant, $F$ and $F_{ext}$ are the two-dimensional controlled and external forces respectively, and $u$ is the two-dimensional control vector. The time constant of the linear muscle model was set to $\tau = 60$ms [93]. The parameters $m$ and $G$ were arbitrarily set to 1 to standardize the simulations. To simulate the presence of a mechanical perturbation to the system we changed the value of $F_{ext}$ to the amplitude of the perturbation. This perturbation amplitude will propagate to the subsequent timesteps thanks to Eq 3.

The system dynamics was discretized using Euler method with a time step of 10ms to integrate stochasticity, which gave the following difference equation

$$x_{t+1} = Ax_t + Bu_t + \xi_t \tag{4}$$

where $x_t$ is the time-dependent state vector containing position, velocity, controlled and external forces, $A$ and $B$ are two real-valued matrices that capture system dynamics, $u_t$ is the command vector at time $t$, and $\xi_t \sim N(0, \Sigma_m)$ is additive motor noise defined by a multivariate normal distribution of zero mean and covariance matrix $\Sigma_m$. To include the goal target, we augmented the state vector with another vector of the same dimension (dimension 16 in total, it facilitates simulations of tasks in which any subset of state variables has to be penalized). Finally, the state vector was further augmented ($\tilde{x}_t$) to integrate feedback delays and let the controller observe only the most delayed state [31,94]. The corresponding feedback signals at each time step write

$$\begin{aligned} y_t &= \tilde{x}_{t-h} + \omega_t \\ &= H\tilde{x}_t + \omega_t \end{aligned} \tag{5}$$

where $h$ is the feedback delay, $\omega_t \sim N(0, \Sigma_s)$ is additive, zero-mean sensory noise with covariance matrix $\Sigma_s$ and $H = [0^{16x16}, \ldots, 0^{16x16}, I^{16x16}]^T$ is the observability matrix. The feedback delay was set to 50ms (h = 5) to capture the sensory motor delay associated with long-latency feedback responses, assuming that this transcortical loop supports goal-directed, state-feedback control [39]. Because of the noise and sensory delays that corrupt the sensory feedback, we used a dynamic Bayesian estimator weighting sensory feedbacks and priors to estimate the state vector. The maximum likelihood estimator of the posterior distribution computed through a Kalman filter is denoted $\hat{x}_t$ and follows

$$\hat{x}_{t+1} = A\hat{x}_t + Bu_t + K_t(y_t - H\hat{x}_t) \tag{6}$$

where $K_t$ are the Kalman gains recursively computed (see [27] for more details). The optimal motor commands are computed by minimizing a cost function consisting of a weighted sum of quadratic penalties on the state vector (first term) and on the motor command (second term) which writes as follows

$$J(\boldsymbol{x}_{1,\ldots,N}, \boldsymbol{u}_{1,\ldots,N}) = \sum_{i=1}^{N} (x_i^T Q_i x_i + u_i^T R u_i) \tag{7}$$

where $Q_i \geq 0$ and $R > 0$ are real-valued matrices respectively capturing the penalties on the state vector and motor command, $\boldsymbol{x}_{1,\ldots,N}$ and $\boldsymbol{u}_{1,\ldots,N}$ capture the successive state and command vectors, and $N$ is the number of time steps (i.e. time horizon). In most previous studies as well as in the present one, $N$ is an integer which characterizes a finite horizon formulation of the control problem. Under these assumptions, the optimal motor commands $u_t^*$ are defined by a

linear combination of the different entries of the state estimate

$$u_t^* = -L_t \hat{x}_t \tag{8}$$

where $L_t$ are the optimal feedback gains computed recursively (see [27] for more details).

## Online changes in target structure

The optimal feedback control framework allows to determine the optimal control policy for a given set of task parameters $\Theta = \{Q, R, N\}$, capturing respectively the structure of goal target, the impact of motor cost, and the remaining movement duration. The control policy obtained this way is specifically tuned to these task parameters and can be written at each time step as follows

$$\pi^\Theta : u_t^* = -L_t^\Theta \hat{x}_t \tag{9}$$

where the superscript $\Theta$ indicates that this control policy has been computed for the set of task parameters $\Theta$. The optimal feedback gains $L_t^\Theta$ are computed offline and applied during movement execution. If we consider a second set of task parameters $\tilde{\Theta} = \{\tilde{Q}, R, N\}$ that only differs by the penalty on the state vector $\tilde{Q}$, it will define an optimal control policy $\pi^{\tilde{\Theta}}$ such that $\pi^{\tilde{\Theta}} \neq \pi^\Theta$. This principle has been applied in previous work to model trial-to-trial variation in task demands captured for instance by target of different shapes or different penalties to state variables [2,95]. As such, this framework does not model movements in which the set of task parameters $\Theta$, used to determine the control policy, changed during movement. However, recent behavioral experiments have demonstrated humans' ability to optimally adjust their control policy during movements following dynamic changes in target structure [17,18].

In the case of time-varying task requirements (e.g. a change in target structure during movement), captured by a time-varying sets of task parameters $\Theta$, we propose a recursive dynamical computation of the motor commands that considers, at any time during movement, the set of task parameters available to the controller. The basic premise of this implementation is that the set of optimal feedback control gains $L_t^{\Theta(t)}$ are computed at each time step from the time-varying set of task parameters $\Theta(t) = \{Q(t), R, N\}$ and only the first motor command is applied, similarly to what is done in the MPC framework [47]. The procedure is then repeated at the next time step, for which $Q(t)$ and $N$ are adapted to respectively capture time-varying task demands and decrease in remaining movement duration. The remaining movement duration (N) was decremented at each time step such that the total movement time remained constant. This recursive implementation of the optimal feedback gains is schematically represented in Fig 7A, in the more general framework of OFC.

In order to validate this implementation, we simulated two different tasks that we previously conducted experimentally. The first task consisted of reaching movement towards a goal initially presented as a narrow or wide target, the main axis of which being orthogonal to the main reaching direction. During movement, the target width could instantaneously switch from narrow to wide or vice versa [17]. The state vector penalty matrices (first term in Eq (10)) for these two targets were defined such that the following equalities were verified, respectively for the narrow and wide targets

$$J_1(x_N) = w_1 (p_y(N) - p_y^*)^2 + w_1 (p_x(N) - p_x^*)^2 + (\dot{p}(N) - \dot{p}^*)^2 \tag{10}$$

$$J_1(x_N) = w_1 (p_y(N) - p_y^*)^2 + w_2 * (p_x(N) - p_x^*)^2 + (\dot{p}(N) - \dot{p}^*)^2 \tag{11}$$

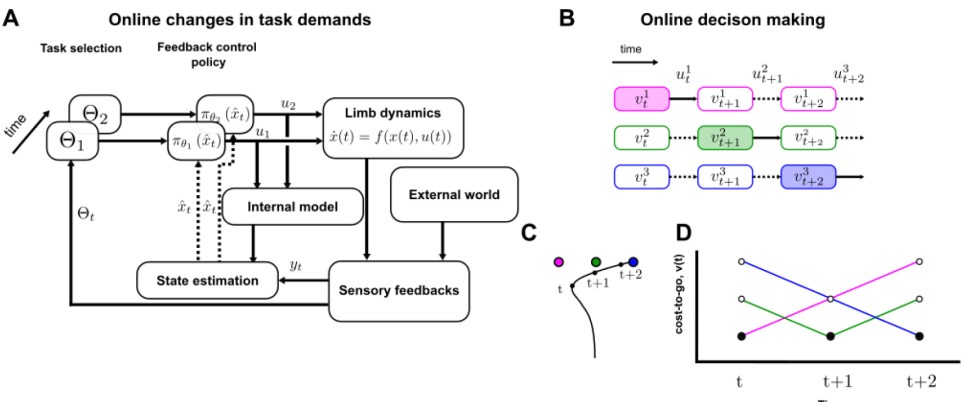

**Fig 7. Model implementation. A** Schematic representation of the computation of online changes in task demands. At each time step, the time-varying task parameters $\Theta_t$ are used to derive the optimal control policy $\pi_{\Theta_1}(\hat{x}_t)$ that is used to compute the motor command $u_t$ (black arrow) which depends on the dynamical state estimate $\hat{x}_t$ (dotted arrow) computed through dynamical Bayesian integration (state estimation). **B** Schematic representation of the implementation of online motor decisions in a three targets paradigm (see panel **C**). Each line schematized the time-varying cost-to-go functions associated with each option which are compared such that the one associated with the lowest value (see panel **D**) is selected (represented by the filled rectangle) and the corresponding motor command (full black arrow) is applied to the system for that very time step. **C** Representation of the different targets and an exemplar simulated hand trajectory (exaggerated case for illustration), the dots correspond to the time at which the decisions processes were considered. **D** Cost-to-go functions associated with the three targets evaluated at three different time points. The filled dots represent the minimum values that instructed the decision process, whenever these filled dots fall on a new color, it corresponds to an online change in target.

where the $^*$ superscript refers to the target state, $p(t)$ is the position at time t, the subscripts refer to the x- and y-coordinates, $w_1$ and $w_2 = 0$ are the weights associated with the narrow and wide dimensions of the targets. This means that there were no constraints on the end-point x-position of the wide target. The different gain values were arbitrarily selected such that the reaching behavior was qualitatively similar to what is observed experimentally. In addition to the changes in target structure, we induced mechanical step perturbations during movement by setting the value $F_{ext} = 5N$ for the x-coordinate of this force.

The second task that we simulated to validate this implementation was the one presented in [18], which was similar to the previous one except that the target width could continuously change during movement. To model these phenomena, we defined the following time-varying state penalty matrices for the fast and slow conditions

$$J_1(x_N) = w_1(p_y(N) - p_y^*)^2 + w_2^{slow,fast}(t)(p_x(N) - p_x^*)^2 + (\dot{p}(N) - \dot{p}^*)^2 \tag{12}$$

$$w_2^{slow}(t) = \frac{0.001 * w_2}{1 + exp\left(\frac{10-t}{3}\right)} \tag{13}$$

$$w_2^{fast}(t) = \frac{0.01 * w_2}{1 + exp\left(\frac{10-t}{3}\right)} \tag{14}$$

where $w_2 = 100$. The time-dependency of these two parameters captures the continuous changes in target width.

To match the experimental findings that reported a delay of about 150ms between the change in target structure and the onset of differences observed in the muscle activity [17,18], we introduced a hard delay of 150ms (15 time-steps) to every modulation of target structure.

Importantly, this delay is different from the one associated with the long-latency feedback responses elicited by mechanical perturbations [39,40] as it requires to perform an additional update of the control policy [17,50].

## Reward-dependent changes of mind

The classical formulation of the Optimal Feedback Control considers a single goal target which is used to derive the set of optimal feedback gains $L_t$. In order to consider multiple potential targets during movement, Nashed and colleagues [25] proposed to leverage the cost-to-go function that appears in the dynamic programming resolution of the control problem [96]. In this resolution, the optimal feedback gains are computed recursively thanks to the following backward recursion

$$L_t = (R + B^T S_t B)^{-1} (B^T S_t A) \tag{15}$$

$$S_{t-1} = Q_t + A^T S_t (A - B L_t) \tag{16}$$

$$s_t = s_{t-1} + tr(S_t \Sigma_m) \tag{17}$$

$$S_N = Q_N, s_N = 0 \tag{18}$$

where $S_t$ are real-valued matrices that capture the instantaneous penalty on the state vector, $s_t$ is a scalar value involved in the computation of the cost-to-go function, and $tr(.)$ denotes the trace operator. Under the optimal policy $\pi$, we define the estimated cost-to-go function as the remaining expected cost if the optimal policy is followed from the initial state $x_t$ to the target state $x^*$, this function writes as follows

$$\hat{v}^\pi(x_t) = \hat{x}_t^T S_t \hat{x}_t + s_t \tag{19}$$

This expression of the cost-to-go can be derived from the Dynamic Programming method by relying on Bellman equation which recursively defines the cost to go as follows

$$\hat{v}^\pi(x_t) = x_t^T Q_t x_t + \pi(\hat{x}_t)^T R \pi(\hat{x}_t) + E[\hat{v}^\pi(x_{t+1})]$$

Where E is the expectation operator. The proof is available in other studies [97,98]. This cost-to-go function is tightly related to the notion of state-value function in the theory underlying reinforcement learning, as both problems rely on the Bellman equation to formalize their resolution [57,58]. The system dynamics we considered in the present study does not involve signal-dependent noise for clarity (see Eq (4)), but we verified that this simplification did not affect the modeling results as it impacts the cost-to-go function independently of the modification we suggested (see Eq 11 in [98]). We therefore decided to only consider additive noise in the present work, simulations for the signal-dependent noise are shown in appendix (see Fig A in S1 Text).

In theory, the cost-to-go is defined based on the true state vector $x_t$ and on the impact of noise in the system, captured by the parameter $s_t$. Here, however, the true state vector is not known from the point of view of the controller. Thus, because the best guess about the state is $\hat{x}$, it is natural to consider that the best guess for the cost-to-go is the form in Eq (19). Note that the zero-mean Gaussian variable that is the error between the true and estimated state can be factored out of the first term of Eq (19), leading to an estimated cost-to-go equal to the true cost-to-go plus an error-term that influences $s_t$. Similar developments apply if signal-dependent noise is considered, where error-related terms in the cost-to-go are already present.

To integrate multiple target states or alternative options for the movement, we computed the cost-to-go associated with each of these options based on the current state estimate, similarly to what Nashed and colleagues proposed [25]. In this study, all targets had the same reward; however, recent experimental reports demonstrated that reward could bias online motor decisions towards solutions that were not optimal if only sensorimotor factors were taken in isolation [20,21,23,53]. Thus, it was necessary to alter the terminal value of the target to include these reward-related biases ($s_N$).

Here we included explicit reward into the OFC formalism to allow reward-dependent biases in target selection as observed experimentally. In principle, Eq (19) represents an estimate of the total expected cost to accumulate from the current time step to the movement horizon. Clearly, an external reward can be taken in consideration by offsetting the value $s_t$, which has a meaningful impact on the cost-to-go while leaving the recurrence and controller unchanged. Indeed, the updated backward recursion and corresponding cost-to-go function writes as follows for a reward bias $r_N$ on the goal target.

$$L_t = (R + B^T S_t B)^{-1}(B^T S_t A)$$

$$S_{t-1} = Q_t + A^T S_t(A - BL_t)$$

$$s_t = s_{t-1} + tr(S_t \Sigma_m)$$

$$S_N = Q_N, s_N = r_N$$

$$\hat{v}^\pi(x_t) = \hat{x}_t^T S_t \hat{x}_t + s_t$$

The impact of this reward bias will therefore only affect the offset of the cost-to-go function without altering the feedback gains.

To simulate online motor decisions between alternative options, we computed the set of $S_t^i, L_t^i$ and $s_t^i$ associated with each option $i$. Then, at each time step, the cost-to-go functions $v^i$ associated with each option $i$ are computed and compared, including rewards that were not related to the ongoing movement throughout the terms $s_t^i$, and the motor command associated with the smallest cost-to-go function amongst all the options is selected. Observe that if $v^i < v^j$ for some time, and then suddenly $v^j < v^i$ at some time, the corresponding control law ($u^j$ instead of $u^i$) effectively implements an online change in target. This implementation is novel in that it proposes a control framework to simulate reaching movements in presence of multiple targets potentially associated with different rewards. The basic premise is that, during movement, the controller continuously keeps track of the cost-to-go associated with each option by combining sensorimotor and cognitive factors and select the optimal action by comparing these different cost-to-go values. A schematic representation of this implementation is proposed in Fig 7B–7D. Together, our implementation consists in a single feedback control policy that selects actions according to the current state of the body and environment to optimally solve the control problem.

We validated this implementation by comparing its predictions with experimental data collected in three experiments. The first experiment was similar to that reported in our previous study, the controller had to reach to any of three potential targets [21]. For each target, we used the state penalty defined in Eq (10) and considered two different reward distributions: (i) all the targets had the same reward ($s_0 = 0$) or (ii) the central target had a higher reward ($s_0 = 5.10^{-2}$) than the other two ($s_0 = 0$). The second and third experiments that we simulated to validate our model investigate reaching movements directed towards a redundant rectangular

target that had a non-uniform reward distribution along its redundant axis, similarly to these used in [20,23]. In the second experiment, we considered three different reward distributions along the redundant axis: (i) a symmetric distribution with higher rewards on both ends captured by Eq (20) or an asymmetric distribution biased to the (ii) left or (iii) right, respectively captured by Eqs (21) and (22).

$$s_0^1(x) = -10x^2 + 0.025 \tag{20}$$

$$s_0^2(x) = \begin{cases} -15x^2 + 0.0375, & \text{if } x < 0 \\ -5x^2 + 0.0375, & \text{if } x \geq 0 \end{cases} \tag{21}$$

$$s_0^3(x) = \begin{cases} -5x^2 + 0.0375, & \text{if } x < 0 \\ -15x^2 + 0.0375, & \text{if } x \geq 0 \end{cases} \tag{22}$$

where $x$ is the horizontal position (in cm) along the redundant axis, centered on zero. In all the simulations, we selected the reward values such that they could bias the cost-to-go function. Therefore, the selected values for $s_0$ were of similar magnitude as the standard cost-to-go values.

## Data analysis

Besides the simulation and characterization of human's behavior in the various tasks mentioned above, we also used some of our previous experimental data to investigate whether the modulation of decision time predicted by our model was also present in the behavioral data. For this purpose, we investigated our previously collected dataset [99] in which participants performed reaching movements towards multiple targets in presence of explicit target reward and perturbations of different magnitudes [21]. More specifically, we investigated for a given reward distribution and perturbation amplitude the lateral hand deviation along the transverse axis to determine the onset of difference between the trials that reached the central and the lateral targets.

To extract the decision time, we used receiver operator curves (ROC) to determine the onset of change of mind in the kinematics. Briefly, the ROC curves compute the probability that two signals could be discriminated by an ideal observer. We compared trials with the same perturbation, the same reward distribution but with different final targets reached. The area under the ROC curve was calculated for the distribution of trials across participants and the onset of target-specific response was identified when the ROC curve exceeded 0.75 [100,101]. The time reported in the Results section corresponds to that onset of target-specific differences.

## Dryad DOI

https://doi.org/10.5061/dryad.79cnp5hx8 [99]

## Supporting information

**S1 Text. Simulations in presence of multiplicative noise.**
(DOCX)

## Acknowledgments

We thank Hari Kalidindi and Eric Wang for useful suggestions and comments on earlier versions of this manuscript.

## Author Contributions

**Conceptualization:** Antoine De Comite, Frédéric Crevecoeur.

**Formal analysis:** Antoine De Comite, Philippe Lefèvre, Frédéric Crevecoeur.

**Investigation:** Antoine De Comite.

**Software:** Antoine De Comite.

**Writing – original draft:** Antoine De Comite, Philippe Lefèvre, Frédéric Crevecoeur.

**Writing – review & editing:** Antoine De Comite, Philippe Lefèvre, Frédéric Crevecoeur.

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
