## [Decision Letter · Decision Letter 0]

11 Apr 2023

Dear Dr De Comite,

Thank you very much for submitting your manuscript "Continuous monitoring of cost-to-go for flexible reaching control and online decisions" for consideration at PLOS Computational Biology.

As with all papers reviewed by the journal, your manuscript was reviewed by members of the editorial board and by several independent reviewers. The reviewers were generally positive about the paper and the ability of the model to account for experimental observations. However, the reviewers raised several concerns. Most importantly, the reviewers felt that there was insufficient discussion of conceptually related preceding literature. In particular, recent papers from Vassileios Chritopoulos (e.g. Enachescu et al., PLOS CB 2021). I would also add a paper by Emanuel Todorov ("Compositionality of optimal control laws", NIPS 2009). These papers both propose models in which a variable analogous to the cost-to-go is used to arbitrate between multiple policies during the course of a movement. (see also Rigoux and Guigon, PLOS CB 2012). In addition, the reviewers raised some issues about the formulation of the model (including whether model predictive control is the appropriate framework) and whether or not alternative hypotheses could be adequately ruled out.

In light of the reviews (below this email), we would like to invite the resubmission of a significantly-revised version that better situates the current work in the context of previous literature examining the inter-relation of decision making and control (the reviewers provide many examples). The revisions should, of course, also take into account the reviewers' other comments.

We cannot make any decision about publication until we have seen the revised manuscript and your response to the reviewers' comments. Your revised manuscript is also likely to be sent to reviewers for further evaluation.

Sincerely,

Adrian M Haith

Academic Editor

PLOS Computational Biology

Thomas Serre

Section Editor

PLOS Computational Biology

Reviewer's Responses to Questions

**Comments to the Authors:**

Reviewer #1: Here the authors propose a computational model that uses a higher-level decision-making controller that chooses at each time point the minimal cost-to-go from several parallel plans. The cost on accuracy is manipulated to accommodate target width, target size rate of change, and reward. Using this model, they can replicate hand trajectories from a number of previously published results. The paper was an enjoyable read, the model and simulations were well described, and the manuscript is worthy of publication. Below are mostly high-level comments to further improve the paper. Can the authors please respond in a point-by-point manner and refer to line numbers with manuscript changes.

Specific Concerns

1. This first point is probably the most important to address in the paper. Since its introduction to the field, optimal feedback control has been one of the most powerful frameworks to understand human behavior. What has been powerful about this approach is that a lot of behaviour emerges from the model. Here, however, the authors manipulate the cost function differently for each new experimental condition in an ad hoc fashion---rather than all the behaviour emerging from the same model, with parameters and cost function held constant, given task demands. In other words, applying a new cost function for each experimental condition in an ad hoc manner is less powerful than having behavior emerge from the same model. Can the authors please include this as a limitation within the discussion.

2. The intersection of sensorimotor control and decision-making has been modelled previously, but these are noticeably not discussed in the paper and worthy of a discussion paragraph. Some of these citations are below.

• Haith, A. M., Huberdeau, D. M., & Krakauer, J. W. (2015). Hedging your bets: intermediate movements as optimal behavior in the context of an incomplete decision. PLoS computational biology, 11(3), e1004171.

• Christopoulos, V., Bonaiuto, J., & Andersen, R. A. (2015). A biologically plausible computational theory for value integration and action selection in decisions with competing alternatives. PLoS computational biology, 11(3), e1004104.

• Enachescu, V., Schrater, P., Schaal, S., & Christopoulos, V. (2021). Action planning and control under uncertainty emerge through a desirability-driven competition between parallel encoding motor plans. PLoS computational biology, 17(10), e1009429.

Further, these papers and others noted below, suggest competing hypotheses as to whether the sensorimotor system uses a single flexible plan or averages parallel plans. Here the authors provide a slightly different idea. That the nervous system evaluates all parallel plans but can flexibly switch to the plan with the current lowest cost-to-go. Two things to address: 1.) Can the authors please include in the discussion on how their model fits in the literature with other models and hypotheses in the field. 2.) Can the model predict the results of Chapman et al. (2010) or Wong, A. L., & Haith, A. M. (2017) in a go-before-you-know paradigm when the cost-to-go for each potential target is the same.

• Wong, A. L., & Haith, A. M. (2017). Motor planning flexibly optimizes performance under uncertainty about task goals. Nature communications, 8(1), 14624.

• Chapman, C. S., Gallivan, J. P., Wood, D. K., Milne, J. L., Culham, J. C., & Goodale, M. A. (2010). Reaching for the unknown: multiple target encoding and real-time decision-making in a rapid reach task. Cognition, 116(2), 168-176.

• Gallivan, J. P., Logan, L., Wolpert, D. M., & Flanagan, J. R. (2016). Parallel specification of competing sensorimotor control policies for alternative action options. Nature neuroscience, 19(2), 320.

• Alhussein, L., & Smith, M. A. (2021). Motor planning under uncertainty. Elife, 10, e67019.)

Minor Concerns

1. Recent work in both reaching (Mehrabi, N., Sharif Razavian, R., Ghannadi, B., & McPhee, J. (2017). Predictive simulation of reaching moving targets using nonlinear model predictive control. Frontiers in computational neuroscience, 10, 143; Bashford, L., Kobak, D., Diedrichsen, J., & Mehring, C. (2022). Motor skill learning decreases movement variability and increases planning horizon. Journal of Neurophysiology, 127(4), 995-1006.) and gait (Darici, O., & Kuo, A. D. (2022) Humans plan for the near future to walk economically on uneven terrain. arXiv preprint arXiv:2207.11224.) highlight the horizon of planning and or MPC. Can the authors expand in the introduction, with an additional paragraph, the works that have used MPC or considered the horizon of control to understand sensorimotor control in the literature.

2. Last paragraph of the discussion. Decision-making literature, much of the work pioneered by Paul Cisek, suggests that decision-making and control are intertwined. For the past two decades the general thought has been that the nervous system does not plan, decide, act. This would suggest that the cost-to-go is continually monitored, in terms of both energetics and evidence-based decision-making processes. Can the authors please comment and potentially adjust this paragraph.

3. In the decision-making and foraging literature would suggest the cost is reward rate = (reward - energy) / time (Carland, M. A., Thura, D., & Cisek, P. (2019). The urge to decide and act: implications for brain function and dysfunction. The Neuroscientist, 25(5), 491-511). Here rewards are added at a higher decision-making level. It is plausible that rewards themselves influence feedback gains. Can the authors please comment, and perhaps add to the paragraph currently in the discussion.

4. A classic prediction from Fitt’s law is that participants move faster when reaching to big targets than small targets. Do the previous data set find that wider targets led to faster movement? Are movement times constrained in these experiments? If so, what would happen with forward velocity if movement times were not constrained? In the paper the authors manipulate only the x-dimension of the cost function to manipulate target size by manipulating the position state cost. I assume that this does not lead the model to predict faster forward velocity when reaching towards larger targets. Can the authors please comment and include in the discussion.

5. Paragraph 442-452. The ideas of the influence on movement speed due to reward or due to environmental uncertainty are shown to influence forward velocity. But these two are not the same underlying mechanism. The real question is how reward influences the feedback gains, and how environmental uncertainty influences feedback gains, which are likely very different mechanisms. Also, I know the authors have used the word vigor here to discuss feedback gains / forward velocity in the past. However, in general, vigor is more colloquially known in the field as a product of reward.

6. line 105-108: Can the authors expand on how decision-times predicted from their model would differ from diffusion models.

7. line 198. Use another phrase than ‘thanks to’.

Sincerely,

Joshua Cashaback

Reviewer #2: This article presents a model which accounts for human motor behavior following perturbations. It is based on optimal feedback control combined with monitoring of the expected cost-to-go function to decide about motor goal. The model explains a broad range of observations collected in previous experiments from the authors and from other research groups.

The article is well written and rather easy to read. The simulations are properly run and the results are convincing.

I have four comments:

1. line 116 "We extended the optimal feedback control framework ...", line 544 "we propose a recursive dynamical computation ...". I am perplex about these statements. In fact, the OFC framework intrinsically embeds these characteristics (Bryson and Ho 1975, Applied Optimal Control; Todorov and Jordan 2002, Nat Neurosci 5:1226). Even if the control low can be precalculated (Eq. 8), it is implicit in the view of Todorov and Jordan that it is recalculated at each timestep to take into account changes in the body and the environment. There is no need to invoke a recursive dynamical computation nor MPC, in particular because MPC involves a receding horizon (moving horizon; fig. 1 in Garcia et al. 1989, Automatica 25:335), which is not the case here (see below). In the same way, monitoring the cost-to-go function is not a novelty (Todorov and Jordan 2002; Todorov 2004, Nat Neurosci 7:907) and has been used for decision making in the framework of motor control (Rigoux and Guigon 2012, PLoS Comput Biol 8:e1002716). I would suggest to clarify these issues.

2. The model involves a 50-ms delay for sensory feedback and a 150-ms delay to update the control policy. The latter delay seems rather adhoc. It could be more conservative to say that there is a single feedback delay corresponding to the time it takes to detect changes in the body or the environment from sensory information. The idea that it takes time to update the control policy is related to the use of the LQG framework and the fact that the control policy is task-dependent (for a different view, see Guigon 2023, Psychol Rev 130:23).

3. line 98 "We modeled online modifications in control by implementing a receding horizon controller inspired by model predictive control", line 549 "Q(t) and N are adapted to respectively capture time-varying task demands and changes in movement horizon". I see nowhere in the article what is the value of the receding horizon, and whether and how N is actually updated. This is a critical issue for online motor decision between alternative goals. Cost calculation is dependent on intended movement duration. In a normative approach (e.g. Shadmehr et al. 2010, J Neurosci 30:10507; Rigoux and Guigon 2012), a rewarded goal can become more or less interesting depending on the intended time to reach the goal (Stevens et al. 2005, Curr Biol 15:1855). The proposed model seems to bypass this issue.

4. As a minor point, I would suggest, for each simulation result, to add a pointer to corresponding figures in the corresponding experimental articles. For instance the simulation results in Fig. 1 are related to experimental results in De Comite et al. (2021).

Reviewer #3: This paper by Comite, Lefevre, and Crevecoeur proposed a switching mechanism of the cost-to-go during the optimal feedback control problem in humans and animals. During goal-directed reach movements tasks where the requested movement is slow and not ballistic, this proposed architecture is able to switch the optimal control policy in response to the change of the task goal. This model replicated the number of previously reported behavioral results. The proposed hypothesis is modestly new, and the proposed formulation of the model is solid. However, I have several concerns that I would like the authors to address before this gets published. Especially, the method section has to be more detailed and additional simulation studies to explain the characteristics of this model are necessary.

Major concerns

1) Since OFC based model is very flexible in that the setting of the cost function determines its generated behavior, it can explain any behavioral characteristics by manually tuning the cost function for the simulation. In other words, the computational model based on OFC often lacks falsifiability. Thus, setting an alternative hypothesis and refuting this alternative possibility is crucial for a scientific paper. In this paper, the author explains the simulation results for explaining the previous reports straightforwardly without setting an alternative hypothesis explicitly. One alternative hypothesis they discussed is the target switching hypothesis, which predicts the fixed response time. (although I think this prediction is too specific, which is not always true for the switching hypothesis). For example, we can think about another possibility, such as the continuous update of the target representation in the brain is modulated by the reward information. This hypothesis also seems to be derived from previous studies since continuous target remapping based on Bayesian integration has been proposed and discussed in the motor learning literature (e.g., S Vaziri JNS 2006). My intention here is not to propose a specific alternative hypothesis. Rather, I recommend that the authors consider several potential hypotheses from the previous literature and then formalize them as computational models. Model comparisons of multiple models of alternative hypotheses make this paper much stronger.

2) The most important new concept in the proposed computational model has not been well described in the main section. For instance, Figure 6, schematic diagrams of this model, was referred to in the method. To make the scientific argument clear, this should appear in Figure 1 ( main or intro) instead of Figure 6 (method)

3) L 407, “the cost-to-go is monitored continuously” The word monitor is not appropriate here since the selection mechanism of the different cost-to-go was not proposed and not examined. A possible mechanism is, for example, competition among different cost-to-g. In this case, nothing monitors cost-to-go, and one option is automatically and spontaneously selected as a result of the competition. Instead, “the hypothesis that the different cost-to-go are computed continuously” is appropriate.

4) The model ( Eqn(4) ) does not have signal dependent nose. The signal-dependent noise and the smooth minimum variance trajectory is the most important clue of the OFC theory. These SDN should largely influence the simulated behavior. I recommend the author incorporate SDN into the model. If the proposed model for predictive cost-to-go switching architecture can’t accommodate SDN, the model should not be plausible.

5) Most of the idea of the proposed computational model has already been proposed in Nashed(25). Although there is an original contribution of this present paper, such as the offset of the cost-to-go by reward, the main argument of this paper (online switch of the cost-to-go) has been proposed in Nashed(25). Then, which is the real novelty of the present paper? If the reward bias is the original contribution, the introduction has to be rewritten.

6) How exactly the endpoint reward offset (say, r_b(N)) biases the s_t is unclear. This part is not well written in the method section (l638-l642). Considering the algorithm of the OFC solution, the backward computation of DP (Eqn 16), taking account of this bias, should be needed. A detailed additional explanation using equations is necessary.

7) The equations of the selection policy of the cot-to-go v_i,v_j (L647) should be needed. How does the action selection probability is computed? Was the soft-max-function was used? Then, please discuss the reason why this is rational.

8) There is a gap in the logic between the conventional OFC model (Eqn 15-18) and the RL model-like value estimate in Eqn(19). Please explain more about the reason why the formulation of Eqn(19) was necessary.

9) L264. I did not understand how the perturbation was incorporated into the OFC and how this perturbation altered the cost-to-go. This part should be clearly explained in the method section.

10) L318. The reader would like to understand the reason why RT becomes different in this model. What is the factor that alters RT? Is the estimated hand position matter, the variance of the estimate matter, or both matter? Additional simulations study (sensitivity analysis) is needed here to show the sensitivity of independent variables to the RT. The current explanation is not done well step-by-step.

Minor

1) Please add the plots of the original behavior data, adapting it from the original paper. Otherwise, it is inconvenient to understand the simulation results.

2) L561, Eqn 11, “*” of “100*” is confusing and not accurate as a mathematical notation. Please delete them.

**Have the authors made all data and (if applicable) computational code underlying the findings in their manuscript fully available?**

Reviewer #1: **No: **Model should be publicly available.

Reviewer #2: **No: **The code is not available. The experimental data corresponding to some simulations are not available (Refs 17 and 18) while other are available (Ref 21).

Reviewer #3: None

PLOS authors have the option to publish the peer review history of their article (what does this mean?). If published, this will include your full peer review and any attached files.

Reviewer #1: **Yes: **Joshua G.A. Cashaback

Reviewer #2: **Yes: **Emmanuel Guigon

Reviewer #3: No
---

## [Decision Letter · Decision Letter 1]

14 Jul 2023

Dear Dr De Comite,

Thank you very much for submitting your manuscript "Continuous evaluation of cost-to-go for flexible reaching control and online decisions" for consideration at PLOS Computational Biology.

As with all papers reviewed by the journal, your manuscript was reviewed by members of the editorial board and by several independent reviewers. In light of the reviews (below this email), we would like to invite the resubmission of a significantly-revised version that takes into account the reviewers' comments.

Although the reviewers were mostly satisfied with the responses to the previous review comments, there remain a number of issues to be resolved. Specifically, the reviewers still had concerns about 1) The exact formulation of the model (receding horizon and model-predictive control?)  2) The need to reformulate details of the model to simulate different tasks in an ad-hoc way, and 3) How the findings might be impacted by signal-dependent noise.

Of these I believe that 1) still needs to be resolved. I agree with Reviewer 2 that the presentation of the model is unclear and I am similarly confused about how any aspect of the model relates to model-predictive control or a receding horizon. Is it just the idea that the optimal policy (or policies) are recomputed at each timestep? Line 95-97 introduce the model in terms of model-predictive control and receding-horizon control, but this seems to mismatches with the description of the implementation details.

Regarding points 2) and 3), I think these are important caveats to the current contribution, and should be clearly stated as such in the paper, but I don't view them as critical weaknesses. Ad-hoc tuning of models is a general issue with optimal control models, but this is fairly orthogonal to the main idea conveyed in the work (i.e. that the decision to switch between targets varies in a stereotyped way and this can be accounted for by the model). Similarly, while signal-dependent noise does provide a strictly more realistic model of the sensorimotor system, and there would certainly be additional challenges in extending these ideas to a signal-dependent noise setting (as Reviewer 3 highlights), the core conceptual idea can still be applied and I don't think the core insights from the LQG setting are invalidated.

Therefore, in your revision, please address the issues surrounding the confusion with model presentation, and ensure that the discussion adequately addresses the limitations and possible future extensions of the approach.

We cannot make any decision about publication until we have seen the revised manuscript and your response to the reviewers' comments. Your revised manuscript is also likely to be sent to reviewers for further evaluation.

Sincerely,

Adrian M Haith

Academic Editor

PLOS Computational Biology

Thomas Serre

Section Editor

PLOS Computational Biology

Reviewer's Responses to Questions

**Comments to the Authors:**

Reviewer #1: The authors have made several changes that have improved the manuscript. I just have a couple remaining comments. Please see latest comments below that are denoted, "LATEST COMMENT"

Specific concerns

1. This first point is probably the most important to address in the paper. Since its introduction to the field, optimal feedback control has been one of the most powerful frameworks to understand human behavior. What has been powerful about this approach is that a lot of behaviour emerges from the model. Here, however, the authors manipulate the cost function differently for each new experimental condition in an ad hoc fashion---rather than all the behaviour emerging from the same model, with parameters and cost function held constant, given task demands. In other words, applying a new cost function for each experimental condition in an ad hoc manner is less powerful than having behavior emerge from the same model. Can the authors please include this as a limitation within the discussion.

We understand the Reviewer’s concern and discuss that point in more detail in the revised version of the manuscript. A single parametrization of the optimal feedback control model can indeed result in different behavioral realizations. Reaching behaviors in presence of perturbations (visual or mechanical) and the selective corrections for the effect of noise can be generated using the same cost-function as the one used to generate unperturbed movements (Todorov & Jordan, 2002). However, alterations of the target structure (Knill et al., 2011; Nashed et al., 2012), adaptation mechanisms to velocity-dependent force fields (Crevecoeur et al., 2020; Ikegami et al., 2021) or multiple potential targets (Nashed et al., 2014) necessitate adjustments of the controller, either by changing the cost-function to accommodate specific task requirement, or by changing the model parameters to represent adaptation. The set of tasks that we considered in the present study fall in the category where sudden changes of goal or target structure requires considering the respective cost-function of each configuration or option. The specific contribution of this paper is to demonstrate that the within-movement corrections observed experimentally can be predicted by updating the control policy online thanks to a continuous calculation of the cost-to-go. We clarified this point in the introduction (lines 86-87) and discussion sections (lines 415-417).

“Even though the OFC framework has been extensively used to explain flexible feedback control (2,29–32), it was mainly applied to static environments: that is, cost parameters have been typically considered fixed during a given trial. Therefore, a mechanism able to respond to these sudden changes in task parameters as observed in previous experimental studies is required.”

“To accommodate the experimental observations made about the dependency of these changes, we proposed an extended implementation of the OFC framework combined with a decision module based on a continuous observation of the target structure and an estimation of the cost-to-go function. The addition of this decision module to the native OFC implementation is necessary as it is often assumed that the cost function, and therefore the feedback gains, are fixed during movement, which was not compatible with the experimental results captured in this study.”

LATEST COMMENT: The responses above, while helpful, I do not believe fully address this concern---which is also the related to the first major concern by reviewer 3 (and while the response there is also helpful, I would argue that this concern is not fully addressed and still needs to be mentioned as a limitation). Specifically, the theory / model becomes unfalsifiable when the cost function (independent of task dynamics) is changed for each unique experimental condition, which the authors have done here. Here the authors must hand tune the cost function for each experimental condition (e.g., dot vs bar, slow vs. fast, etc.). That is, there are as many degrees of freedom (by hand tuning) as there are experimental conditions, so it is not possible to falsify. An alternative would be that between / within trial adaptation or perhaps some other time-dependent process naturally changes the cost function, but that is not done in this paper.

Conversely, a model is particularly strong if it does not need to be tuned for different / unique experimental designs, and the behaviour simply emerges from the model for all conditions.

Based on this, I would encourage that the authors include this specific concern as a limitation within in the discussion. Perhaps, simply stating: “A limitation of our approach is that the cost function was manipulated for each experimental condition, rather than predictions emerging for each condition using a model with a constant set of parameters.”

2.) “By introducing the cost-to-go as a decision variable, we bridged an important gap between current models of motor control and decision making. We used the cost-to-go as a decision variable because of its theoretical grounding in the dynamic programming solution of the optimal control problem (54). It has been used to derive locally and globally optimal solutions to the control problem through stochastic optimal control (27,55) and reinforcement learning (56,57). The implementation presented in this paper allows to go beyond standard static formulation of OFC models with fixed parameters, and explicitly integrate the reward, changes in target structure and multiple targets in the cost-to-go, similarly to what was done with the value function (i.e. the opposite of the cost-to-go) in reinforcement learning (58). It is important to mention previous studies investigating how humans performed reaching movements in a paradigm where two or more targets were presented prior to movement, and the correct one was cued later (59–62). The behavior observed in this go-before-you-know paradigm can be reproduced by averaging the control policies or target locations associated with the different options (62–65), simulations not shown. Thus, from a computational perspective this problem does not require to update the controller online which differs from the experimental results that we modelled. In addition, in our task, no target was cued before or during movements, the controller must decide between different valid choices instead of waiting to see which one is correct. However, a controller endowed with dynamic updates based on continuous tracking of the cost-to-go can also handle the problem of reaching to uncertain targets.”

LATEST COMMENT: The authors have included simulations for a go-before-you-know task in the rebuttal and mentioned them in the discussion. From their response above, it seems that they used a model without continuous monitoring (which they mention is not necessary). They also state in the updated manuscript that a controller with dynamic updates based on continuous tracking of the cost-to-go can handle reaching to uncertain target.

Coming back to the Wong and Haith (2017) paper, a critical experimental condition they have not captured with their rebuttal simulations is the fast condition. Specifically, humans will not select an ‘average’ trajectory when there is a heavy time constraint, but rather select the left or right target by moving directly towards that target (see Figure 2D, bottom row with blue trajectories in the Wong and haith paper). The authors’ model would not make this prediction given its current formulation. It is understood that the focus of the paper is online changes, but it is also important to clearly highlight what aspects of the decision-making / motor control literature that their model can or cannot explain.

Can the authors please make a more direct statement that the model in its current formulation does not capture all of the results in the Wong and Haith (2017) and Alhussein and Smith (2021). Perhaps, “In the current formulation of our model, we are not able to capture all the experimental conditions in the Wong and Haith (2017) or Alhusseign and Smith (2021). However, a controller endowed with…”

As a minor comment, I would remove the word endowed in their currently written sentence and replace with another.

Sincerely,

Josh Cashaback

Reviewer #2: The authors have done a great job answering every single question of the 3 reviewers. Yet I am still completely confused about the receding horizon. Does the model involve a receding horizon?

lines 95-97

"We modeled online modifications in control by implementing a receding horizon controller inspired by model predictive control (33,34), which was previously used to study sensorimotor control (35–37)"

lines 575-576

"N is considered to be a finite integer which characterizes a finite horizon formulation of the control problem"

At this stage, I understand that N is given at the beginning of the simulation and is decremented at each time step.

line 584

N is now in the set of task parameters so maybe it means that N can be changed.

line 602-605

"The basic premise of this implementation is that the set of optimal feedback control gains Lt are computed at each time step from the time-varying set of task parameters theta(t) and only the first motor command is applied, similarly to what is done in the MPC framework. The procedure is then repeated at the next time step, for which Q and N are adapted to respectively capture time-varying task demands and changes in movement horizon. The time horizon (N) was decremented at each time step such that the total movement time remained constant."

I do not understand. Horizon can be changed during the movement but you manage so that movement time is constant. The logic escapes me.

Note again that the fact that "only the first motor command is applied" is not specific to MPC but is also in the definition of optimal feedback control (Bryson and Ho 1975, p 128).

From wikipedia (https://en.wikipedia.org/wiki/Model_predictive_control)

"MPC is based on iterative, finite-horizon optimization of a plant model. At time t the current plant state is sampled and a cost minimizing control strategy is computed (via a numerical minimization algorithm) for a relatively short time horizon in the future: [t,t+T]. Specifically, an online or on-the-fly calculation is used to explore state trajectories that emanate from the current state and find (via the solution of Euler–Lagrange equations) a cost-minimizing control strategy until time t+T. Only the first step of the control strategy is implemented, then the plant state is sampled again and the calculations are repeated starting from the new current state, yielding a new control and new predicted state path. The prediction horizon keeps being shifted forward and for this reason MPC is also called receding horizon control. Although this approach is not optimal, in practice it has given very good results. Much academic research has been done to find fast methods of solution of Euler–Lagrange type equations, to understand the global stability properties of MPC's local optimization, and in general to improve the MPC method."

Things should not be complicated. There are two approaches (forget the infinite horizon formulation):

1. Fixed horizon

Movement duration T is given in advance. The number of steps is N = T/timestep. The algorithm is the following: for each n in [0,N], optimize over [n,N]. This is the classical optimal feedback control used by Todorov and Jordan.

2. Receding horizon

Simulation (and not movement) duration T is given in advance and a receding horizon TH is defined. The number of simulation steps is N = T/timestep and the number of steps of the horizon is NH = TH/timestep. The algorithm is the following: for each n in [0,N-NH], optimize over [n,n+NH].

There is a third approach that we could call "variable horizon":

Movement duration T is given in advance. The number of steps is N = T/timestep. The algorithm is the following: for each n in [0,N], optimize over [n,N], decide to change N (or not) for some reason (perturbation) and use the new N.

What approach is actually used in the model? This is a fundamental question. When you want to compare baseline and perturbed trials in a motor task, the two types of trial will generally lead to different movement durations. A fixed horizon is not a very reasonable way to address perturbations. An adhoc solution is to choose movement duration based on experimental observations (e.g. Liu and Todorov 2007, J Neurosci 27:9354). A principled (but maybe wrong) approach is to use a receding horizon.

Emmanuel Guigon

Reviewer #3: I appreciate the revisions made to the manuscript in accordance with all reviewers' comments. Thanks to the additional information on the model and the task, my understanding of the proposed model has improved.

However, I have concerns about the generality of the proposed model and its arguments. It appears they are confined only to the Linear Quadratic Gaussian (LQG) framework and cannot be extended to a more biologically plausible model of the optimal feedback control, as proposed by Todorov in 2002.

The fundamental question of how the brain handles signal-dependent noise is the starting point of this theoretical framework. The flexibility in the variance of ongoing movement can only be explained by the optimal feedback control framework, which is capable of incorporating signal-dependent noise in sensory and motor signals. The proposed algorithm is built on dynamic programming and consists of multiple iterations of feedforward steps for state estimation and backward steps for policy optimization. Importantly, these two processes are not independent in this algorithm. Once the cost-to-go is set, the algorithm must compute the entire optimal sequence of state variables (x_hat(t)) to achieve optimal feedback gain over time (see page 1092 of Todorov's 'Stochastic Optimal Control and Estimation Methods Adapted to the Noise Characteristics of the Sensorimotor System', Neural Computation 2005).

Thus, if there are two task parameters for the cost-to-go (Theta_1 and Theta_2) as shown in Fig 7, each optimal feedback control (OFC) component computes a different feedback policy (L_1, L_2) for the different time series of state variables (x_hat1(t=1:end), x_hat2(t=1:end)). Consequently, if the Cost-to-go switches, for instance from 1 to 2 at time t=t_switch, the optimal policy L_2(t_switch) optimizes the sequence of x_hat2(t_switch) rather than the actual state generated by L_1 (i.e., x_hat1). Therefore, if it exists in the biological system as Todorov predicted, the optimal feedback control system would need to recompute the optimal L_2(t=t_swith:end) and x_2(t=t_swithc:end) with the new constraint of the initial state xhat_2_new(t=t_swich)=x_1(t=tswich), which is different from the original x_hat_2.

This is a crucial aspect that computational neuroscientists need to address to demonstrate the validity of the proposed integration of task selection and the optimal feedback model. If they ignore signal-dependent noise, they can follow the conventional LQG framework. In this case, thanks to the separation of the controller and the estimator, they can avoid this problem. The authors chose not to address this pivotal issue by stating, "We indeed chose to present our model without signal dependent noise in the paper for simplicity." Although they claimed to have replicated the results with signal-dependent noise, I remain skeptical about how accurately this model was derived and operated given the aforementioned reason. I also disagree with the authors' argument that oversimplification is a reasonable excuse for glossing over this significant aspect. If this issue is not resolved in this model (it is not mentioned at all), I believe that the contribution of this paper is quite limited.

I strongly recommend that the authors revise all simulations in this paper using a model that incorporates signal-dependent noise, and explicitly state how they address this problem in Figure 7.

The proposed model is very similar to the existing model of online switching of feedback control policy by the observed target information (target ball’s state ) “Ronsse, Renaud, Kunlin Wei, and Dagmar Sternad. "Optimal control of a hybrid rhythmic-discrete task: the bouncing ball revisited." Journal of Neurophysiology 103.5 (2010): 2482-2493.”.

While it should be appreciated that the author explained some reported data by their group by using the existing task switching architecture of the conventional OFC model without SDN, the novelty and generality of this paper is very limited.

**Have the authors made all data and (if applicable) computational code underlying the findings in their manuscript fully available?**

Reviewer #1: Yes

Reviewer #2: **No: **Already explained

Reviewer #3: **No: **

PLOS authors have the option to publish the peer review history of their article (what does this mean?). If published, this will include your full peer review and any attached files.

Reviewer #1: **Yes: **Joshua Cashaback

Reviewer #2: **Yes: **Emmanuel Guigon

Reviewer #3: No
---

## [Editor Report · Decision Letter 2]

5 Sep 2023

Dear Dr De Comite,

We are pleased to inform you that your manuscript 'Continuous evaluation of cost-to-go for flexible reaching control and online decisions' has been provisionally accepted for publication in PLOS Computational Biology.

Best regards,

Adrian M Haith

Academic Editor

PLOS Computational Biology

Thomas Serre

Section Editor

PLOS Computational Biology

---

## [Editor Report · Acceptance letter]

22 Sep 2023

PCOMPBIOL-D-23-00171R2 

Continuous evaluation of cost-to-go for flexible reaching control and online decisions

Dear Dr De Comite,

I am pleased to inform you that your manuscript has been formally accepted for publication in PLOS Computational Biology. Your manuscript is now with our production department and you will be notified of the publication date in due course.

With kind regards,

Anita Estes
